# Role of the TRPC1 Channel in Hippocampal Long-Term Depression and in Spatial Memory Extinction

**DOI:** 10.3390/ijms21051712

**Published:** 2020-03-03

**Authors:** Xavier Yerna, Olivier Schakman, Ikram Ratbi, Anna Kreis, Sophie Lepannetier, Marie de Clippele, Younès Achouri, Nicolas Tajeddine, Fadel Tissir, Roberta Gualdani, Philippe Gailly

**Affiliations:** 1Université Catholique de Louvain, Institute of Neuroscience, Cell Physiology, av. Mounier 53/B1.53.17, B-1200 Brussels, Belgium; xavier.yerna@uclouvain.be (X.Y.); olivier.schakman@uclouvain.be (O.S.); Ikram.ratbi@uclouvain.be (I.R.); slepannetier@gmail.com (S.L.); marie.declippele@uclouvain.be (M.d.C.); nicolas.tajeddine@uclouvain.be (N.T.); roberta.gualdani@uclouvain.be (R.G.); 2Université Catholique de Louvain, de Duve Institute, Transgenic Core Facility, av. Hippocrate 75/B1.75.09, B-1200 Brussels, Belgium; younes.achouri@uclouvain.be; 3Université Catholique de Louvain, Institute of Neuroscience, Developmental Neurobiology, av. Mounier 73/B1.73.16, B-1200 Brussels, Belgium; fadel.tissir@uclouvain.be

**Keywords:** memory extinction, TRPC, long-term depression, mGluR

## Abstract

Group I metabotropic glutamate receptors (mGluR) are involved in various forms of synaptic plasticity that are believed to underlie declarative memory. We previously showed that mGluR5 specifically activates channels containing TRPC1, an isoform of the canonical family of Transient Receptor Potential channels highly expressed in the CA1-3 regions of the hippocampus. Using a tamoxifen-inducible conditional knockout model, we show here that the acute deletion of the *Trpc1* gene alters the extinction of spatial reference memory. mGluR-induced long-term depression, which is partially responsible for memory extinction, was impaired in these mice. Similar results were obtained in vitro and in vivo by inhibiting the channel by its most specific inhibitor, Pico145. Among the numerous known postsynaptic pathways activated by type I mGluR, we observed that the deletion of *Trpc1* impaired the activation of ERK1/2 and the subsequent expression of Arc, an immediate early gene that plays a key role in AMPA receptors endocytosis and subsequent long-term depression.

## 1. Introduction

Canonical Transient Receptor Potential (TRPC) proteins are nonselective cation membrane channels. Seven TRPC isoforms (TRPC1 to TRPC7) have been described, all of which form homo- and/or heterotetrameric Ca^2+^-permeable channels. TRPC channels have diverse functions in the brain, such as neurite outgrowth and axon guidance, neural progenitor cell proliferation and differentiation, and neuronal apoptosis or survival [1,2,3]. TRPC channels are also able to modulate neuronal excitability and could promote excitotoxicity [4,5]. Among these isoforms, TRPC1 is the most widely distributed. It is particularly highly expressed in the hippocampus and in the amygdala. In contrast to other isoforms, TRPC1 does not seem to form homomers in adult neurons but it can heterotetramerize with TRPC4 and/or TRPC5 [6,7].

Several activation mechanisms have been proposed for TRPC channels, the most frequently described being the activation by G-protein-coupled receptors or by receptor tyrosine kinases via the phospholipase C (PLC)-induced formation of diacylglycerol and inositol 1, 4, 5-trisphosphate [8,9,10]. TRPC1 in particular can also be involved in store-operated calcium entry in cooperation with Orai1 channel and activated by STIM1, a sensor of Ca^2+^ contents in the endoplasmic reticulum [11,12,13,14]. It can be directly activated by phosphatidylinositol 3,4,5-trisphosphate, by phosphorylation by the protein kinase C (PKC), or through the hydrolysis of phosphatidylinositol 4,5-bisphosphate and/or the concomitant generation of diacylglycerol [15] or other lipids, such as sphingosine-1-P [16]. In neurons, we and others reported that TRPC1 is activated by type I metabotropic glutamate receptors mGluR1 or mGluR5 [17,18,19]. These receptors are coupled to PLC via G_q_ and G_11_ G proteins. Recently, Myeong and colleagues showed that heterotetrameric TRPC1/C4 and TRPC1/C5 channels were activated by a direct interaction with activated Gα_q_ [20] and were subsequently inhibited by the PLC β-induced phosphatidylinositol-4,5-bisphosphate depletion, the release of Ca^2+^ from the reticulum and the activation of PKC [21].

As type I mGluRs play a crucial role in synaptic plasticity [22,23,24,25], we studied the possible role of TRPC1 in the hippocampus, one of the most important area of the brain required for the formation and the retrieval of various types of memory [26]. The following two forms of synaptic plasticity are commonly believed to underlie learning and memory processes: the long-term potentiation (LTP) and the long-term depression (LTD) corresponding to the ability of synapses to strengthen or weaken over time, in response to their pattern of activity. These processes have been largely studied in the Schaffer collateral-commissural pathway between the CA3 and CA1 regions of the hippocampus and in the mossy fibres’ pathway between the dentate gyrus and CA3 pyramidal cells. It is widely assumed that LTP and LTD reflect synaptic and cellular processes that occur during formation/storage and the extinction of memories, respectively. Both processes depend on a rise of intracellular Ca^2+^ during synaptic activity [27]. We recently showed that the activation of TRPC1 channels by type I mGluRs transiently depolarizes hippocampal neurons in culture and modifies their excitability. In vivo, the lack of TRPC1 channel impairs these processes and results in a functional impairment of theta-burst stimulation-induced LTP and of spatial working memory [18].

The direct chemical stimulation of group I mGluRs with dihydroxyphenylglycine (DHPG) actually weakens synapses (reviewed in [22,28,29,30]). This process underlies some forms of plasticity obtained after a low frequency pattern of stimulation. It has been largely studied but its functional significance is only partially understood. It could reverse a beforehand reinforced synapse and correspond to a “depotentiation” or weakens naive synapses and correspond to a real LTD [31]. Depotentiation could explain the erasure of synaptic memory trace and be considered as a forgetting mechanism. LTD, as a premise of synapse elimination might be required to refine and consolidate memories. In the present paper, these processes are further indistinctly referred to mGluR-LTD. It is now commonly admitted that mGluR-dependent synaptic plasticity plays a key role in the reversal learning and in cognitive flexibility, allowing a suppression of previously acquired memory and acquisition of new information to adapt to novel situations [32,33,34]. For example, mGluR5 knockout mice exhibit only minor deficits in the rate of acquisition of reference spatial memory but have significant deficits in reversal task [35].

In the present paper, we investigated the role of TRPC1-containing channels in memory extinction and its most likely cellular correlate, the LTD induced by mGluR stimulation. As TRPC1 also interferes with memory acquisition (in particular in spatial working memory and in fear conditioning, see [18]), we used two strategies—one pharmacological, the other genetic—to impair TRPC1 function just before or just after the learning process and evaluated its role in memory extinction.

## 2. Results

### 2.1. Trpc1 Conditional Knockout Mouse Model

We previously showed that *Trpc1*^−/−^ mice present alterations in spatial working memory and in fear conditioning. However, using this complete knockout mouse model, it was not possible to address in vivo the role of TRPC1 specifically in neurons, as the expression of TRPC1 is also impaired in glial cells. It was also not possible to exclude a developmental impairment due to TRPC1 depletion during embryonic development. This is of particular importance as TRPC1 has been reported to interfere with neural stem cells proliferation. In order to control the expression of TRPC1 specifically in neurons and at a specific time point, we therefore built a conditional knockout model in which the exon 2 of the *Trpc1* gene was flanked by lox recombination sites (Figure 1). *Trpc1^lox/^*^−^ mice were crossed with mice expressing the CreERT2 fusion protein under control of the regulatory elements of the Ca^2+^/calmodulin-dependent protein kinase IIα gene (*CaMKII-CreERT2* transgene) [36,37]. Transgenic *Trpc1^lox/^*^−^
*CaMKII^Cre^*^−/−^ mice (hereinafter referred as *CTRL*) and their littermates *Trpc1^lox/^*^−^
*CaMKII^Cre+^*^/−^ (hereinafter referred as *cKO*) were injected intraperitoneally twice a day for 5 days with 1 mg of tamoxifen. After a delay of 10 days, we observed that the expression of TRPC1 measured by RT-qPCR was diminished by about 70% in *cKO* (Figure 1).

### 2.2. Acute Deletion of the Trpc1 Gene Impairs Spatial Memory Extinction

The Morris Water Maze (MWM) has been designed to assess spatial reference memory. In this test, mice progressively learn during five consecutive days to locate a hidden platform using visual cues. The performance is evaluated by measuring the time to reach the platform (escape latency). On the 5^th^ day of the experiment, the platform is removed and the spatial reference memory is evaluated by measuring the proportion of time spent in the quadrant where the platform had been present earlier (probe test). We previously showed that reference spatial memory is normal in *Trpc1*^−/−^ mice compared to WT but that *Trpc1*^−/−^ mice learn slightly more slowly, presenting a longer escape latency at the beginning of the process (day 2, see Figure 2G in [18]). Here, to repress the expression of TRPC1 specifically in neurons and to do so acutely at three months of age and therefore avoid possible developmental impairments, we compared memory performances of *cKO* to *CTRL* mice injected intraperitoneally twice a day for 5 days with 1 mg of tamoxifen. An MWM assay was performed after a delay of 10 days, i.e., at a time point where the expression of *Trpc1* was significantly decreased (Figure 1B). As expected, we observed that acute deletion of the *Trpc1* gene did not affect the acquisition of reference spatial memory. Indeed, both *cKO* and *CTRL* mice treated with tamoxifen reached the platform in about 50 s on the first trial (day 1), then, the escape latency decreased from day to day similarly in both strains (Figure 2A). The swimming speed was also similar, suggesting the absence of any major locomotion defect. Memory extinction was then evaluated by measuring during five consecutive days, the proportion of time the mice spent in the quadrant from which the platform had been removed (repeated probe tests). An important extinction could be observed in *CTRL* mice (treated with tamoxifen) but it was significantly impaired in *cKO* mice (also treated with tamoxifen, Figure 2B), suggesting that TRPC1 is involved in the extinction of spatial reference memory. In order to investigate the role of TRPC1 channel in memory extinction and to avoid any interference with a possible implication in memory acquisition, we genetically impaired TRPC1 expression just after the learning process. The mice were submitted to MWM assay and were thereafter injected intraperitoneally twice a day for 5 days with 1 mg of tamoxifen. Reference memory extinction was evaluated after an additional delay of 10 days, i.e., three weeks after the first trial in the MWM. As expected, the initial reference memory was similar in the two groups, but as shown in Figure 2C, memory extinction was significantly impaired in tamoxifen-injected *cKO* mice compared to *CTRL,* again pointing out the role of TRPC1 in the extinction process.

### 2.3. Pharmacological Inhibition of TRPC1/4/5 Channels Impairs Spatial Memory Extinction

Memory extinction was similarly measured by repeated probe tests in MWM after the pharmacological inhibition of TRPC1. Pico 145 is the most specific inhibitor of this TRPC isoforms. It inhibits TRPC1/TRPC4 and TRPC1/TRPC5 heterotetramers with an IC50 around 10 and 100 pM, respectively, and TRPC4 and TRPC5 homotetramers with an IC50 at least 10 times higher [38]. It is not active on other TRP channels and on a wide range of molecular targets tested, including ion channels, receptors, and kinases [39]. Pico 145 was used at 0.1 mg kg^−1^ weight, i.e., at a dose 10 × smaller than the one previously used in vivo for TRPC4 and 5 inhibition [40]. Mice were trained in a water maze from day 1 to day 5. Memory extinction was studied by repeated probe tests from day 8 to day 12. During this period, one hour before each probe test, the mice were injected intraperitoneally with 0.1 mg kg^−1^ weight Pico145 or with vehicle. As shown in Figure 3, memory extinction was significantly impaired in the group injected with Pico145.

### 2.4. Inhibition of TRPC1 Channels Affects mGluR-Induced LTD

Agonists of group I mGluRs are kown to induce a chemical LTD of synaptic transmission in the CA1 region. As they activate TRPC1, we investigated the possible involvement of TRPC1 in LTD and compared mGluR-dependent LTD chemically evoked with 50 μM DHPG in brain slices from *CTRL* and *cKO* mice injected with tamoxifen. Schaffer collaterals were stimulated and field excitatory postsynaptic potentials (fEPSP) were recorded in the stratum radiatum of the CA1 region. The relationship between the stimulus intensity and the fEPSP slope was similar in slices from both genotypes (Figure 4A). In *CTRL* animals, DHPG induced a reduction of the fEPSP slope down to 25% of the initial value, which then stabilized at around 70% of the pre-stimulation level. As shown in Figure 4B, LTD was reduced by half in brain slices from *cKO* mice. The involvement of TRPC1 in mGluR-LTD was also tested pharmacologically. We observed that after pre-incubation of the slice with 1 nM Pico145, mGluR-LTD was almost completely inhibited (Figure 4B).

### 2.5. Deletion of the Trpc1 Gene Impairs the Type I mGluR-Induced Signaling Pathway

We previously showed that neurons isolated from *Trpc1*^−/−^ mice or from *cKO* mice treated for 48h with 1 µM OH-tamoxifen present significantly reduced [Ca^2+^]_i_ transients in response to DHPG. These observations suggested that the acute and specific neuronal inhibition of TRPC1 was sufficient to decrease type I mGluR-induced Ca^2+^ entry. Here, we studied the signaling pathways downstream mGluR involved in LTD. As the results concerning DHPG-induced Ca^2+^ entries and DHPG-induced LTD were similar in *cKO* mice injected with tamoxifen, in mice treated with pico145, and in *Trpc1*^−/−^ mice, experiments were conducted on this latter model. Brain slices from *Trpc1^+/+^* and *Trpc1*^−/−^ mice were stimulated with 50 µM DHPG for 0, 5, 20, or 60 min. We first observed that, in WT slices, the expression of Arc protein significantly increased immediately after a stimulation of 5 min (Figure 5A). This effect was not observed in brain slices from *Trpc1*^−/−^ animals. Among the factors susceptible to induce Arc transcription and translation, we measured the phosphorylation and activation of the extracellular signal-regulated kinases 1/2 (ERK1/2). In WT slices, we observed a 3 to 4 times increased phosphorylation ERK1/2, the time course of which fitted with Arc expression, i.e., a phosphorylation occurring at 5 and 20 min. Recently, another important pathway in Arc translation has been discovered, including calpain-1, a calcium-dependent protease and B56α, the regulatory subunit of PP2A phosphatase (Figure 5B). We measured the expression of spectrin, a known substrate of calpain-1 and of B56α. A very minor decrease in the expression was observed after 60 min. This effect was not seen in slices from *Trpc1*^−/−^ animals. The expression of GluA1 and GluA2 subunits of AMPA receptors did not change significantly following DHPG stimulation.

## 3. Discussion

In the present paper, we demonstrate that TRPC1 is required for spatial memory extinction. This process is important in behavioral flexibility, as it allows new learning when a relevant spatial information is no longer valid. We previously showed that TRPC1 is activated by mGluR5 [18]. On the other hand, it is known that mGluR5 is required for the memory extinction that occurs in the absence of a context change but not in a new context [32,34]. Interestingly, both the acute inhibition of mGluR5 by 2-methyl-6-(phenylethynyl) pyridine hydrochloride (MPEP, [34]) or the acute inhibition or genetic deletion of TRPC1 (this study) alter spatial memory extinction (evaluated by repeated probes tests in MWM) but have no effect on the acquisition of reference memory following water maze training [41]. Genetically modified mice, where key molecules required for the expression of mGluR-LTD are removed (such as p38 MAPK/MAPK-activated protein kinase 2 and/or 3), have also been shown to present mGluR-LTD impairment associated with deficits in the hippocampal-dependent reversal learning with no significant deficit in the initial learning and acquisition phases [33,42]. Here, we show that both the genetic and pharmacological inhibition of TRPC1 alter mGluR-LTD.

A major discovery in the field has been that mGluR-LTD requires a rapid dendritic protein synthesis [43]. Among the different proteins involved, the activity-regulated cytoskeletal-associated protein Arc is of special interest. Arc is specifically required for long-term memory formation as well as experience-dependent plasticity in circuits (reviewed in [44,45]). Arc is an immediate early gene, the mRNA of which is rapidly transcribed and targeted to dendrites [46]. It is one of the most strongly induced immediate early genes in response to novel environment exposure. The increase of Arc mRNA in specific dendritic spines “primes” them for mGluR-LTD. Thereafter, in response to mGluR1/5 stimulation, Arc is translated very rapidly, i.e., within minutes after stimulation with glutamate, locally in the dendritic spine [47]. The Arc protein then interacts with dynamin and the endocytic machinery that triggers α-amino-3-hydroxy-5-methyl-4-isoxazolepropionic acid receptors (AMPARs) endocytosis and allows late phase of LTD [48]. We observed that in control hippocampal slices stimulated with DHPG, Arc expression increases early, after 5 min of DHPG stimulation. Here, the priming of specific dendritic spines was presumably obtained by standard handling procedures necessary for brain slices experimentation. As such, the DHPG-LTD studied here might in fact be a form of depotentiation instead of a real LTD, as explained in the Introduction [31]. Interestingly, in *Trpc1*^−/−^ mice, the DHPG-induced expression of Arc was not induced. This correlates with the significant diminution observed in the late phase of LTD. We therefore investigated the signaling pathways known to trigger Arc translation in response to mGluR1/5 stimulation, in particular ERK1/2.

Several postsynaptic signaling pathways have been found to be involved in mGluR-LTD. These include the MAPKs, in particular p38 MAPK and ERK1/2 that can be stimulated by G-protein release via the small GTPase Rap1 through the mitogen-activated kinase kinases MKK3/6 and MEK1/2, respectively. ERK1/2 can also be stimulated by group I mGluR-induced activation of the epidermal growth factor (EGF) receptor tyrosine kinase via G protein-release, which activates a cascade including the nonreceptor tyrosine kinase Src. Both p38 MAPK and ERK1/2 activate transcription factors such as Elk-1, CREB, and NF-κB. Moreover, ERK1/2 (but not p38 MAPK) mediates the formation of the translation initiation complex eIF4F complex necessary for initiating protein synthesis [49] (reviewed in [29]). Several proteins are specifically synthesized during mGluR-LTD, including Arc (see above), FRMP, the microtubule-associated protein 1B (MAP1B), and striatal-enriched tyrosine phosphatase STEP that all regulate AMPAR synaptic trafficking.

In our experiments, we observed that ERK1/2 were phosphorylated after 5 min of stimulation with DHPG. The process was completely blunted in brain slices from *Trpc1*^−/−^ mice, suggesting that TRPC1-containing channels that are activated by type I mGluR improve ERK1/2 phosphorylation and activation. Further investigation is needed to understand the molecular mechanisms involved but it is interesting to note that a similar observation was done in non neuronal cells in response to EGFR stimulation [14], suggesting that the EGF-src pathway might be upregulated by TRPC1 activation. These results suggest that TRPC1 exerts its effects in the postsynaptic terminal. Indeed, TRPC1 is directly activated by mGluRs, which are expressed exclusively on dendritic spines outside the postsynaptic membrane specializations (i.e., perisynaptically) [50,51] and this activation modifies known postsynaptic pathways. However, a presynaptic role has also been demonstrated [52].

Recently, the group of Baudry showed that mGluR-LTD depended on calpain-1 [53]. This calcium-dependent protease degrades the B56α subunit of PP2A phosphatase, which decreases PP2A activity and results in the stimulation of the Akt-mTOR-p70S6K pathway, leading to an increase in Arc synthesis and to a reduction in GluA1 expression. We previously showed that, in muscle cells, the influx of calcium through TRPC1-containing channels activates calpain-1 [13] and Akt [54]. We therefore investigated this pathway but observed that the DHPG-induced degradation of B56α and spectrin occurred late (60 min after stimulation) and was only slightly reduced in *Trpc1*^−/−^ mice. Akt was not activated in our conditions (not shown), suggesting a very minor implication of TRPC1, if any, in that pathway. Besides, the total expression of GluA1 and GluA2 was not significantly reduced. This is in line with pioneer studies showing that surface expression but not total expression of GluA1 is reduced after treatment with DHPG [55].

In conclusion, the present paper shows that inhibiting TRPC1 in the hippocampus impairs type I mGluR-induced ERK1/2 activation and subsequent Arc expression, which in turn diminishes mGluR-LTD. This correlates with an alteration of reversal spatial learning. Type I mGluR-LTD occurs in many other brain regions, including the neocortex, midbrain, striatum, and cerebellum (for review, see [56]). We hope that a better understanding of the mechanisms involved in the process will help in treating states in which memory extinction and mGluR-LTD are slowed down, such as in ageing [57], Alzheimer’s disease [58], or in pathologies in which these processes seem exaggerated, such as in Fragile X syndrome and autism [59,60].

## 4. Materials and Methods

### 4.1. Animals

All animals were housed and handled in accordance with European guidelines and were approved by the animal ethics committee of the Université catholique de Louvain (2017/UCL/MD/013, June 2017–May 2021). This study was performed on male 3–4-month-old mice. At appropriate experimental time points, all animals were anesthetized and sacrificed.

### 4.2. Generation of Trpc1 Conditional Knockout Mice

The generation of *Trpc1*^−/−^ mice has been described previously [18]. Briefly, we built mice having the second exon of the *Trpc1* gene flanked with *loxP* site. These mice were bred with PGK-Cre recombinase mice line to obtain a constitutive *Trpc1* knockout mouse line. Heterozygous mice were further bred to obtain homozygous mice on a mixed genetic C57BL6/129S1/Sv background. Heterozygous transgenic mice and their WT littermates were identified by PCR genotyping. Mice with the *lox Trpc1* allele were bred with *Trpc1*^−/−^ mice to obtain the heterozygous *Trpc1^lox/^*^−^ genotype. *Trpc1^lox^* mice were bred with ROSA *stop-lox-stop-Tomato* Cre reporter mice to generate the respective genotypes used for the experiments. In order to achieve temporally controlled somatic mutagenesis specifically in neurons of the forebrain region, the *Trpc1^lox^*^/−^ mice were crossed with mice expressing the CreERT2 fusion protein under the control of the regulatory elements of the *CaMKII*α gene (*CaMKII-CreERT2* transgene) [36,37]. Heterozygous transgenic mice *Trpc1^lox/^*^−^
*CaMKII^Cre+/^*^−^ and their *Trpc1^lox/^*^−^
*CaMKII^Cre^*^−/−^ littermates were identified by PCR genotyping. Transgenic animals were injected intraperitoneally with 1 mg of tamoxifen twice a day for five consecutive days.

### 4.3. Behavioral Assays

The MWM was used to assess spatial learning and memory. The water maze was made of a round pool with a diameter of 113 cm virtually divided into four quadrants (North, South, West, and East) and was filled with water (26 °C). Several visual cues were placed around the pool and a platform was placed at the center of the North-East quadrant of the pool. The time latency to reach the platform, the swim speed, and the time spent in each quadrant were measured. In order to measure spatial memory extinction, mice were tested for four consecutive days by placing them for 1 min in the water maze from which the platform had been removed. The proportion of time spent in each the quadrant was measured.

### 4.4. Brain Slice Preparation

Animals were killed (by cervical dislocation) and their brains were quickly removed and placed in ice-cold artificial cerebrospinal fluid (ACSF) composed of (mM): 126 NaCl, 3 KCl, 2.4 CaCl_2_, 1.3 MgCl_2_, 1.24 NaH_2_PO_4_, 26 NaHCO_3_, 10 glucose (bubbled with 95%O_2–_5% CO_2_%). Slices from the ventral part of the hippocampus were used as this part of the hippocampus has a greater ability to exhibit DHPG-induced LTD. The cerebellum and frontal cortex were removed. The brains were mounted onto a LeicaVT1200 vibratome, and horizontal sections of a thickness of 350 µm were cut in ice-cold ACSF to obtain the ventral hippocampus. The slices recovered in oxygenated ACSF at 32 °C for at least 1h before use.

### 4.5. Field Potential Recordings

Mouse brain slices were transferred to the recording chamber and continuously perfused with oxygenated ACSF (2 mL/min) at 30 °C. Excitatory postsynaptic potentials (EPSP) were evoked through a bipolar stimulating electrode that was placed in the Schaffer collaterals and recorded by the AxoClamp 2B (Axon Instruments, Foster City, CA, USA) amplifier through a glass electrode, which was back-filled with 2M NaCl and placed in the CA1 region (stratum radiatum). The stimuli consisted of 100 µs pulses of constant currents with the intensity adjusted to produce 35% of the maximum response every min. After the placement of the electrodes, responses were stabilized for 30 to 60 min (1 stimulation/min). Responses were digitized by Digidata1322A (Axon Instruments) and recorded to a computer using WinLTP software [61]. LTD was induced chemically by applying 50µM DHPG for 20 min. All LTD experiments were done in the presence of 50 µM picrotoxin and 10 µM D-(-)-2-Amino-5-phosphonopentanoic acid (D-AP5), a potent antagonist of the N-methyl-D-aspartate receptor. Slopes of fEPSP responses were expressed normalized to the pre-treatment baseline, defined as 100%.

### 4.6. RNA Extraction and Real Time qPCR

RNA expression in the hippocampus was evaluated after total RNA was purified using Trizol method. The *Trpc1* primer sequences used were for the forward CAGAAGGACTGTGTGGGCAT and for the reverse CAGGTGCCAATGAACGAGTG. RT-qPCR was performed using 5 µL of cDNA, 12.5 µL of SybrGreen Mix (BioRad, Temse, Belgium), and 250 µM of each primer in a total reaction volume of 25 µL. The reaction was initiated at 95 °C for 3 min and was followed by 40 cycles of denaturation at 95 °C for 10 s, annealing at 60 °C for 1 min, and extension at 72 °C for 10 s. The data were recorded on a DNA Engine Opticon RT-qPCR Detection System (BioRad), and the cycle threshold (Ct) values for each reaction were determined using analytical software from the same manufacturer. To normalize the data, the *gapdh* gene was used as housekeeping gene.

### 4.7. Western Blotting

Brain slices were stimulated with DHPG and were collected (2 slices per sample) in radioimmunoprecipitation assay (RIPA) buffer (25 mM Tris HCl pH 7.6, 150 mM NaCl, 1% NP-40, 1% sodium deoxycholate, 0.1% SDS). The samples were clarified by centrifugation at 14,000× *g* and the protein concentration was determined using a bicinchoninic acid assay kit (BCA; Thermo Scientific, Merelbeke, Belgium). The samples were heated for 10 min at 70 °C. A total of 10 μg of protein for each sample was loaded on a 7% SDS-polyacrylamide gel and transferred to a nitrocellulose membrane. After blocking with 5% non-fat milk, the membranes were probed overnight with anti-Arc (1/1000, Cell Signaling, Leiden, The Netherlands), anti-phospho-p44/42 MPK (pERK1/2, 1/1000, Cell Signaling Technology), anti-spectrin (1/1500, Merck Millipore, Overijse, Belgium), anti-B56α (1/250, Santa Cruz Biotechnology, Dallas, TX, USA), anti-GluA1 (1/500, Merck Millipore), anti-GluA2 (1/1000, Merck Millipore), anti-GAPDH (1/1000, Cell Signaling Technology), or anti-β tubulin (1/1000, Neuromics, Edina, MN, USA). The membranes were then incubated with secondary antibodies coupled to peroxidase (Dako, Agilent Technologies, Diegem, Belgium) and peroxidase was detected with Pierce ECL Plus (Thermo Scientific) on ECL hyperfilm.

### 4.8. Drugs

(RS)-DHPG, picrotoxin and D-AP5 were obtained from Tocris Bioscience (Bristol, UK). Pico145 was generously given by R. Bon and D. Beech (University of Leeds). It was synthesized as previously described and stored at 10 mM in dimethyl sulfoxide [38]. Pico145 was used at a final concentration of 1 nM on brain slices. For in vivo experiments, Pico145 was diluted in PBS buffer and injected intraperitoneally (0.1 mg kg^−1^ weight, final volume injected: 100 µL). All drugs were prepared as stock solutions according to the supplier’s recommendations and stored at −20 °C until use.

### 4.9. Statistical Analyses

The data are expressed as a mean ± standard error of the mean (SEM). For each set of experiments, each *n* represents one slice (for electrophysiology) and one animal (for behavior testing). Statistical significance was assessed by Student’s *t*-test or two-way analysis of variance (ANOVA) tests for comparisons. The statistical significance was fixed to *p* < 0.05.

## Figures and Tables

**Figure 1 ijms-21-01712-f001:**
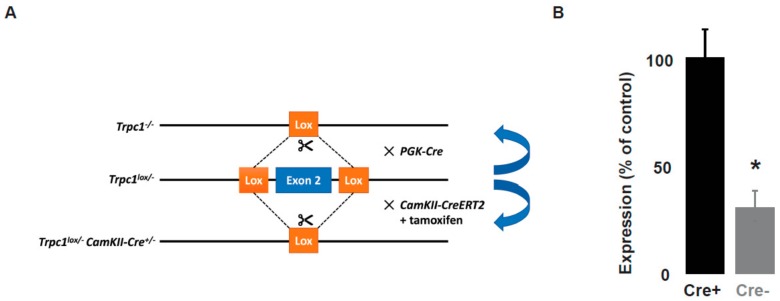
Generation of the *Trpc1* conditional knockout mice. **A**. Mice having the second exon of *Trpc1* gene flanked with *loxP* site (panel A, middle line) were bred either with the PGK-Cre recombinase mice line to obtain a constitutive *Trpc1* knockout mouse line (panel 1, upper line) or with mice expressing the CreERT2 fusion protein under the control of the regulatory elements of the *CaMKII*α gene (*CaMKII-CreERT2* transgene, panel 1, bottom line) to obtain Heterozygous transgenic mice (*Trpc1^lox/^*^−^
*CaMKII^Cre+/^*^−^ mice) (panel A, bottom line). **B**. *Trpc1* expression measured by RT-qPCR analysis in hippocampal samples from *Trpc1^lox/^*^−^
*CaMKII^Cre+/^*^−^ and *Trpc1^lox/^*^−^
*CaMKII^Cre^*^−/−^ mice 10 days after the last IP injection of tamoxifen. *: *p* < 0.05, Student’s *t*-test, *n* = 9 samples of hippocampus from three different mice for each genotype (i.e., 3 independent experiments).

**Figure 2 ijms-21-01712-f002:**
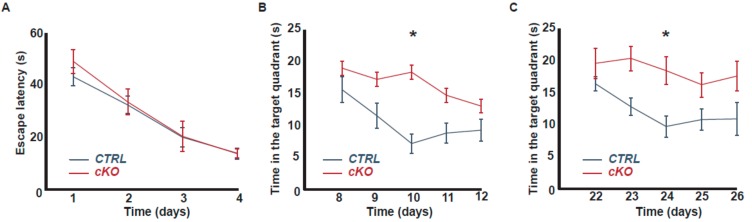
*Trpc1* gene deletion impairs reference memory extinction. Morris Water Maze (MWM) test. **A**. The average escape latencies, i.e., the time required for *CTRL* (blue line, *n* = 15) and *cKO* mice (red line, *n* = 9) to reach the platform. All mice were treated with tamoxifen (5 days injection + 10 days of delay). **B**. Memory extinction evaluated by repeated probe tests (after the platform has been removed) at days 8 to 12. *: *p* < 0.05, two-way repeated measures ANOVA. **C**. The same experiment but the mice were injected with tamoxifen after the learning period, i.e., at days 8 to 12 and memory extinction evaluated at 10 days after, i.e., at days 22 to 26 (*n* = 7 vs 7). *: *p* < 0.05, two-way repeated measures ANOVA.

**Figure 3 ijms-21-01712-f003:**
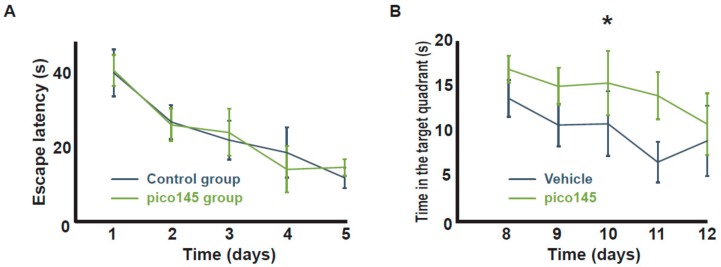
Pharmacological inhibition of TRPC1/4/5 impairs reference memory extinction. *CTRL* mice were submitted to a MWM training and thereafter were injected intraperitoneally with 0.1 mg kg^−1^ weight pico145 (green line, *n* = 6) or with vehicle (blue line, *n* = 5). **A**. The average escape latencies (before injection). **B**. Memory extinction evaluated by repeated probe tests at days 8 to 12, one hour after injection of pico145 or vehicle. *: *p* < 0.05, two-way ANOVA.

**Figure 4 ijms-21-01712-f004:**
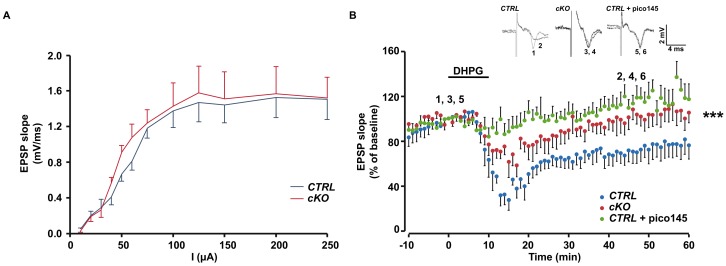
Dihydroxyphenylglycine- (DHPG) induced long-term depression (LTD). **A**. The input–output relationship between field excitatory postsynaptic potentials (fEPSP) measured in CA1 stratum radiatum and the intensity of stimulation in brain slices from *CTRL* mice (blue lines) and *cKO* mice (red lines), both injected with tamoxifen. **B**. Time-course of fEPSP slopes (mean ± SEM) measured before and after stimulation with 50μM DHPG. *CTRL* brain slices (blue dots), *cKO* slices (red dots), and *CTRL* slices treated with 1 nM pico145 (green dots). Upper insets show representative traces of fEPSP before and 40 min after DHPG. ***: *p* < 0.001, one-way repeated measures ANOVA followed by Bonferroni test, *n* = 5 for each group).

**Figure 5 ijms-21-01712-f005:**
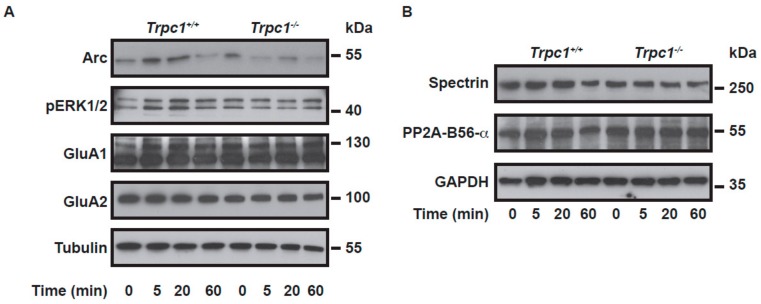
DHPG-induced signaling pathways. Western Blot analysis of protein expression in hippocampal slices stimulated for 0, 5, 20, or 60 min with 50µM DHPG (two slices per sample, *n*= 3 to five independent experiments per group) **A**. The phosphorylation of ERK1/2 and the expression of Arc were significantly increased (by 3.6 times and 1.6 times, respectively) after a 5 and 20 min stimulation with DHPG in the *Trpc1^+/+^* brain slice. **B**. The expression of B56α and spectrin were significantly diminished (by 20 and 22% respectively) after a 60 min stimulation with DHPG in *Trpc1^+/+^* brain slices. These effects were not observed in *Trpc1*^−/−^ mice.

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
