# Peer review of "Role of the TRPC1 Channel in Hippocampal Long-Term Depression and in Spatial Memory Extinction"

_ijms, 2020, doi:10.3390/ijms21051712_

Round 1

Reviewer 1 Report

This work studies the role of TRPC1 channels in different aspects of learning and memory. The performance of conditional KO animals in water maze tests is investigated. The authors conclude that TRPC1 channels are involved in the extinction of spatial reference memory and mGluR mediated LTD. The manuscript is in general clear and well written.

Major points

Electrophysiological evidence of current reduction through TRPC1 channels in hippocampal neurons of conditional KO mice is necessary. This would contribute to improve the shortage of the results shown. The discussion, although well written, is often vague, tedious and unrelated to the results obtained.

Minor Points

P3, L18. The sentence “Indeed ...” is confusing.

P5, L12. The word “expression” is repeated.

Author Response

Dear Editor,

We thank the referees for their positive reactions and critiques. We answer point by point here below.

Ref 1.

Major points :

- Previous electrophysiological measurements have been done on TRPC1+/+ and TRPC1-/- cells. On conditional KO, we used Ca2+ imaging technique, which allowed us to measure 10 to 20 cells at the same time. This is advantageous as TRPC1 repression is not complete and might therefore vary from one cell to another. Both Ca2+ measurements and RT-qPCR lead us to conclude that TRPC1 expression is indeed diminished.

- Discussion has been shortened. Points of the discussion that were not directly related to the results were removed, in particular, all the discussion about the pre- or/and the postsynaptic site of action of TRPC1.

Minor points have been addressed.

Ref 2.

Western blots have been realized for GluA1 and GluA2 subunits of AMPA receptors. As expected (see ref 47), no significant change was observed. This is now included in Fig. 5A and discussed p.5 L.52.

For TRPC1, we did not succeed to detect it correctly with Abcam and Alomone antibodies. In human samples, we did detect it in other studies, but as mentioned in a previous publication, for murine samples, the antibodies used detect a similar signal in WT and KO…! (Tajeddine et al., TRPC1: Subcellular Localization? J Biol Chem (2010) 285)

A series of abbreviations have been removed (DAG, IP3, PIP2, PIP3, DG, SC, TBS, IEG, NMDAR…)

The discussion has been shortened and clarified. Points of the discussion that were not directly related to the results were removed, in particular, all the discussion about the pre- or/and the postsynaptic site of action of TRPC1.

The number of references was also drastically reduced (30 references removed)

The figures and the legends of the figures have been revised.

Reviewer 2 Report

The study linking the TRP channels with glutamate receptor expresion is interesting however in order to confirm the idea I will sugest some major changes:

Please present western blot analysis for the expression of TRPC and glutamate receptors in the different study conditions Please, define abbreviations on their first use along the manuscript and, if possible, try to reduce their use. It is hard to read and follow some paragraphs in which many abbreviations are cited. Try to reduce and clarify the discussion section. The figure legends do not match the figures, I was unable to find any asterisc in the figures in spite of the fact that significant differences are mentioned in the figure legends It seems quite complicate to follow a paper in which there are as many reference pages as introduction, results and discussion together, please try to reduce the number of references. Correct for minor typographic and grammatical mistakes

Author Response

(The authors gave the same response as above.)
